# Ileal Crohn’s Disease Exhibits Reduced Activity of Phospholipase C-β3-Dependent Wnt/β-Catenin Signaling Pathway

**DOI:** 10.3390/cells13110986

**Published:** 2024-06-05

**Authors:** Tomoaki Ando, Ikuo Takazawa, Zachary T. Spencer, Ryoji Ito, Yoshiaki Tomimori, Zbigniew Mikulski, Kenji Matsumoto, Tohru Ishitani, Lee A. Denson, Yu Kawakami, Yuko Kawakami, Jiro Kitaura, Yashi Ahmed, Toshiaki Kawakami

**Affiliations:** 1Laboratory of Allergic Diseases, Center for Autoimmunity and Inflammation, La Jolla, CA 92037, USA; tm-ando@juntendo.ac.jp (T.A.);; 2Atopy Research Center, Graduate School of Medicine, Juntendo University, Tokyo 113-8421, Japan; 3Department of Molecular and Systems Biology and the Dartmouth Cancer Center, Geisel School of Medicine at Dartmouth College, Hanover, NH 03755, USA; zspencer12391@gmail.com (Z.T.S.);; 4Central Institute for Experimental Animals, Kawasaki 210-0821, Kanagawa, Japan; 5Imaging Facility, La Jolla Institute for Immunology, La Jolla, CA 92037, USA; 6Department of Allergy and Clinical Immunology, National Research Institute for Child Health and Development, Tokyo 157-8535, Japan; 7Institute for Molecular and Cellular Regulation, Gunma University, Maebashi 371-0044, Gunma, Japan; 8Division of Gastroenterology, Hepatology, and Nutrition, Cincinnati Children’s Hospital Medical Center, Cincinnati, OH 45229, USA; 9Department of Pediatrics, University of Cincinnati College of Medicine, Cincinnati, OH 45267, USA

**Keywords:** phospholipase C-β3, inflammatory bowel disease, Wnt/β-catenin signaling, KO mouse, Drosophila

## Abstract

Crohn’s disease is a chronic, debilitating, inflammatory bowel disease. Here, we report a critical role of phospholipase C-β3 (PLC-β3) in intestinal homeostasis. In PLC-β3-deficient mice, exposure to oral dextran sodium sulfate induced lethality and severe inflammation in the small intestine. The lethality was due to PLC-β3 deficiency in multiple non-hematopoietic cell types. PLC-β3 deficiency resulted in reduced Wnt/β-catenin signaling, which is essential for homeostasis and the regeneration of the intestinal epithelium. PLC-β3 regulated the Wnt/β-catenin pathway in small intestinal epithelial cells (IECs) at transcriptional, epigenetic, and, potentially, protein–protein interaction levels. PLC-β3-deficient IECs were unable to respond to stimulation by R-spondin 1, an enhancer of Wnt/β-catenin signaling. Reduced expression of PLC-β3 and its signature genes was found in biopsies of patients with ileal Crohn’s disease. PLC-β regulation of Wnt signaling was evolutionally conserved in *Drosophila*. Our data indicate that a reduction in PLC-β3-mediated Wnt/β-catenin signaling contributes to the pathogenesis of ileal Crohn’s disease.

## 1. Introduction

Inflammatory bowel disease (IBD), comprised of Crohn’s disease (CD) and ulcerative colitis, is a complex disease resulting from an interaction between host genetic factors and environmental factors that leads to inflammation in the gastrointestinal tract [1,2]. Heterogeneity and chronicity clinically characterize CD. There are five different types of CD, each affecting different parts of the digestive tract: gastroduodenal CD, jejunoileitis, ileitis, ileocolitis, and Crohn’s colitis. CD may present or evolve with time into more complex clinical manifestations that include strictures, fistulae, and abscesses. CD patients experience highly variable responses to therapies. Potentially related to this complexity, more than 200 genomic loci have been associated with IBD [1,3].

The Wnt family of secretory glycoproteins play important roles in various physiological and pathological processes, including embryogenesis, organ development, tissue homeostasis, metabolism, and tumorigenesis [4,5]. In the absence of Wnt stimulation, cytoplasmic β-catenin is actively phosphorylated, ubiquitinated, and targeted for proteasomal degradation by the β-catenin destruction complex, which includes the tumor suppressors Axin and adenomatous polyposis coli (APC), the Ser/Thr kinases GSK-3 and CK1, the protein phosphatase 2A (PP2A), and the E3-ubiquitin ligase β-TrCP. Wnt proteins interact with Wnt receptor complexes consisting of a member of the Frizzled (Fzd) family and the low-density lipid receptor family members LRP5 or LRP6. Wnt binding to the receptor complexes inactivates the β-catenin destruction complex. Consequently, β-catenin accumulates in the cytoplasm and nucleus and binds to members of the T-cell factor (TCF)/lymphoid enhancer factor (LEF) family transcription factors, leading to the regulation of Wnt target genes. The activation of Fzd1 leads to the activation of Wnt/β-catenin signaling in a Gαq- and Gαo-dependent manner [6], and Fzd5 stimulation by Wnt-5a activates Gαq [7]. The canonical Wnt/β-catenin pathway is essential for intestinal stem cells (ISCs) [4,5,8] and thus for the generation of all intestinal epithelial cells (IECs). Therefore, the dysregulation of this pathway may contribute to IBD pathogenesis [9].

Phospholipase C (PLC) is a family of enzymes that catalyze the hydrolysis of phosphatidylinositol 4,5-bisphosphate to generate two second messengers, i.e., diacylglycerol and inositol 1,4,5-trisphosphate [10]. Thirteen known mammalian PLC isozymes are classified into six subfamilies (PLC-β, PLC-γ, PLC-δ, PLC-ε, PLC-ζ, and PLC-η) [11]. Different PLC isozymes are activated by different receptors and mechanisms, for example, PLC-β is activated by G protein-coupled receptors (GPCRs) and PLC-γ by tyrosine kinase-linked receptors [10]. Notably, both the *PLCB3* gene and a single nucleotide polymorphism (rs792789 allele A) linked to CD and atopic dermatitis maps to human chromosome 11q13 [12,13]. Moreover, atopic dermatitis is more common in patients with IBD than in control populations [14,15]. We previously showed that reduced PLC-β3 expression is associated with a subset of atopic dermatitis [16]. As PLC-β3 is the only one among four isozymes (PLC-β1~β4) that is highly expressed in IECs (Appendix A), we studied a potential role of PLC-β3 in IBD pathogenesis, using a dextran sodium sulfate (DSS)-induced colitis model [17]. Using conventional and conditional *Plcb3* knockout mice, *Plcb3*-deficient IEC lines, and human CD biopsy samples, we found that PLC-β3 positively regulates Wnt/β-catenin signaling. This regulation is conserved from Drosophila to humans. We concluded that the reduced activity of PLC-β3-dependent Wnt/β-catenin signaling contributes to the development of ileal CD.

## 2. Materials and Methods

### 2.1. Study Design

Initially, DSS model experiments were conducted, revealing the differences between WT and *Plcb3^−/−^* mice. Experiments on bone marrow chimeras and conditional KO mice were followed to show the root cause of non-hematopoietic cells for the above differences. Morphological and transcriptomic abnormalities in the intestinal epithelial cells of *Plcb3^−/−^* mice were further confirmed and extended to the discovery of PLC-β3-regulated Wnt/β-catenin signaling and microbiota using epithelial cell mutants, *Plcb3^−/−^* mice, and Crohn’s disease biopsies. PLC-β-regulated Wnt/β-catenin signaling was also shown in Drosophila.

### 2.2. Mice

This study used 8–14-week-old age- and gender-matched mice throughout. Mice were maintained under specific pathogen-free conditions and all the procedures were approved by the Institutional Animal Care and Use Committee of the La Jolla Institute for Immunology (approved protocol AP00001048). *Plcb3 flox* mice were generated by injecting ES cells (MGI: Clone: HEPD0811_7_H11) into mouse embryos, followed by crossing with B6;SJL-Tg(ACTFLPe)9205Dym/J to remove FLP loci and then crossing with B6 to remove Act-FLPe allele. B6.Cg-Tg(Vil1-cre)997Gum/J, STOCK Tg(Col1a2-cre/ERT,-ALPP)7Cpd/J, B6;129-Tg(Cdh5-cre)1Spe/J, B6.Cg-Tg(Itgax-cre)1-1Reiz/J, B6.FVB-Tg(Rorc-cre)1Litt/J, and B6.129P2-Lyz2tm1(cre)Ifo/J mice were obtained from The Jackson Laboratory. B6;129-Tg(Cdh5-cre)1Spe/J mice were backcrossed to B6 mice for at least 8 generations before being used for experiments. For STOCK Tg(Col1a2-cre/ERT,-ALPP)7Cpd/J mice, 75 mg/kg of tamoxifen in corn oil was administered via intraperitoneal injection for 5 consecutive days 28 days before DSS treatment. No statistical methods were used to predetermine the sample size. The genotyping primers used and estimated band sizes are listed below. For *Plcb3* flox genotyping: Forward: 5′-TCCAGATGGTGCCACTCCTT-3′, Reverse: 5′-AGAAGCCATTAGGGTCCACA-3′ (504 bp for WT, 714 bp for flox). For *Plcb3* flox recombined: R: 5′-CACTTCGTTGAGTCTCGGGT-3′ (790 bp with *Plcb3* flox genotyping F for recombined *Plcb3* flox allele).

### 2.3. Drosophila

RNAi lines for *Plc21C* (*Plc21C^i1^*: Vienna Drosophila Resource Center [VDRC] #108395, *Plc21C^i2^*: VDRC #26558, *Plc21C^i3^*: Bloomington Drosophila Stock Center [BDSC] #31269, and *Plc21C^i4^*: BDSC #33719) and *y* (VDRC #106068) were expressed, along with *UAS-Dcr-2* (BDSC #24648), in third instar larval wing imaginal discs using the *hh-Gal4* driver [18]. Crosses were performed at 25 °C.

### 2.4. Study Participants

The patient-based studies were approved by the Institutional Review Board at Cincinnati Children’s Hospital Medical Center (approval number 2011-2285), and consent was obtained from parents and adult subjects and assent from pediatric subjects aged 11 and above. We prepared unstained sections from formalin-fixed, paraffin-embedded, terminal ileal biopsies obtained for clinical care from seven patients with Crohn’s disease (CD) at diagnosis prior to therapy and two non-IBD controls with normal colonoscopy. The two non-IBD controls were aged 15 and 18 years and were both White, of European ancestry, and not Hispanic or Latino. The seven CD patients were mean (range) 11 (8–15) years of age. Five (71%) were female and all were White, of European ancestry, and not Hispanic or Latino. Five (71%) exhibited an inflamed ileum, with four (57%) exhibiting ileal ulcers.

### 2.5. DSS Treatment of Mice

A total of 1–5% (*w*/*v*) DSS was added to drinking water and mice (4–6 mice per group) were treated ad libitum for 5 days. During and after DSS treatment, mice were monitored at least once a day, the body weight was measured every day, and the mice were euthanized according to the guidelines of the US National Institutes of Health and the animal protocols approved by the Institutional Animal Care and Use Committee of La Jolla Institute for Immunology. The results of all mice were included. Stools were evaluated as follows in a genotype-blinded manner. Stool blood: no blood, 0; blood on stool, 1; gross bleeding, 2. Blood samples were analyzed by Hemavet 950 (Drew Scientific, Plantation, FL, USA) to evaluate anemia. For DSS injection experiments, 1 mg of DSS in PBS was intravenously injected once or 40 mg of DSS was orally injected twice a day for 5 days.

### 2.6. Crypt Isolation and Organoid Culture

Small intestines were isolated, cut open, and washed with cold PBS. Villi were removed by scraping and vigorous pipetting. The tissues were chopped into 5 mm pieces and incubated with 2 mM EDTA/PBS for 30 min on ice. After replacing the supernatant with 25 mL of fresh cold PBS, crypts were isolated by pipetting 50 times. The supernatant was filtered through a 70 µm cell strainer, washed with Advanced DMEM/F12, and resuspended in Matrigel before being seeded in culture plates. Advanced DMEM/F12 was supplemented with 100 µg/mL penicillin, 100 U/mL streptomycin, 2 mM GlutaMAX, 1x N-2 supplement, 1x B27 supplement (all from Life Technologies, Carlsbad, CA, USA), 10 mM HEPES, 1 mM N-acetylcysteine (Sigma-Aldrich, St. Louis, MO, USA) supplemented with 7.5% FCS, 100 ng/mL recombinant mouse Noggin (Peprotech, Cranbury, NJ, USA), 100 ng/mL recombinant mouse EGF (Life Technologies), 500 ng/mL recombinant mouse R-spondin 1 (Life Technologies), and Y-27632 (STEMCELL Technologies, Vancouver, BC, Canada) immediately before use. The medium was changed every 2 days and passaged every 7 or 8 days. For passage, dispase was used to dissociate the organoids.

### 2.7. Generation and Validation of Plcb3-Deficient CMT-93 Cell Lines

Three pairs of oligonucleotides (IDT) for sgRNAs targeting mouse *Plcb3* exon 1 (sgRNAs 1 and 2) or 2 (sgRNA 3) which showed high sensitivity and specificity with integrative sgRNA efficacy prediction (iSEP: http://allergy.ims.riken.jp/index.html) or a pair of oligonucleotides for scramble sgRNA were cloned into lentiCRISPRv2 plasmid as previously described [19]. lentiCRISPRv2 was a gift from Feng Zhang (Addgene plasmid # 52961; http://n2t.net/addgene:5296; RRID:Addgene_52961). Briefly, oligonucleotides were annealed and then ligated with T4 ligase (NewEngland Biolabs, Ipswich, MA, USA [NEB]) into lentiCRISPRv2 fragments (lentiCRISPRv2 plasmid was digested with BsmBI (NEB) and then the larger band was gel-purified). Ligated products were transformed into *Stbl3 E. coli* (NEB) and inoculated to LB–agar plates with ampicillin (50 μg/mL) and incubated for 16 h at 37 °C. Colonies were picked up and screened by colony-PCR (1. 98 °C for 2 min, 2. 98 °C for 5 s, 3. 50 °C for 20 s, 4. 72 °C for 30 s, repeat steps 2–4 for 30 cycles, 5. 72 °C for 3 min, 6. 4 °C. The primers are shown below). Thus, chosen plasmids were cultured in 5 mL of LB medium with ampicillin (50 μg/mL) for 16 h. Plasmids were purified with mini-prep (Zymo Research, Irvine, CA, USA) and screened with restriction enzymes (NEB) to confirm the expected band sizes. Purified plasmids were mixed with sequencing primers and sent for Sanger sequencing (GENEWIZ, South Plainfield, NJ, USA) to confirm the insertion of the correct sgRNA sequence. ABI files were analyzed with FinchTV (Geospiza, Inc., Seattle, WA, USA). A total of 100 μL of bacterial glycerol stock with correct plasmids was inoculated into 200 mL of LB medium + ampicillin (50 mg/mL) and cultured for 16 h at 37 °C, and plasmids were purified with midi-prep (MACHEREY-NAGEL, Dueren, Germany). Plasmid concentration was determined by Nanodrop (Therm o Fisher Scientific, Waltham, MA, USA) and confirmed by gel electrophoresis.

*Plcb3*-sgRNA-1: F: 5′-CACCGAACTTACTCCCGCGCCGCA-3′, R: 5′-AAACTGCGGCGCGGGAGTAAGTTC-3′; *Plcb3*-sgRNA-2: F: 5′-CACCGCTTAGCCATGGCGGGCGCG-3′, R: 5′-AAACCGCGCCCGCCATGGCTAAGC-3′; *Plcb3*-sgRNA-3: F: 5′-CACCGAACCTGGTGACCCTGCGTG-3′, R: 5′-AAACCACGCAGGGTCACCAGGTTC-3′; Scramble sgRNA: F: 5′-CACCGGACGCGTGGCGAATCCTACG-3′, R: 5′-AAACCGTAGGATTCGCCACGCGTCC-3′; lentiCRISPRv2 colony-PCR: F: 5′-ACGATACAAGGCTGTTAGAGAGA-3′, R: 5′-GGACTAGCCTTATTTTAACTTGCT-3′

### 2.8. Transfection of CMT-93 Cells and Single-Cell Sorting

CMT-93 cells (ATCC) were cultured in DMEM (Gibco, Grand Island, NY, USA) supplemented with 10% FBS, penicillin/streptomycin, and glutamine (D10 medium). Then, 10 μg of plasmids with sgRNAs were transfected into freshly seeded (before attaching to the bottom of the culture plate) 25% confluent with equivalent numbers of (about 1 × 10^6^ cells) CMT-93 cells with 20 μL of jetPRIME (Polyplus, Illkirch-Graffenstaden, France) and 500 μL of buffer, following the manufacturer’s instructions. Puromycin (6 μg/mL) was added to the culture 48 h after transfection, and cells were cultured for 21 days, changing the medium with puromycin every 3 days. Cells were washed with PBS and then stripped with 2 mL of Trypsin/EDTA by incubating for 20 min; then, 8 mL of D10 medium was added to inactivate trypsin. Cells were centrifuged at 1500 rpm for 5 min and then the supernatants were removed, washed with PBS, and resuspended with FACS sorting buffer (PBS supplemented with 1% FBS, 5 mM EDTA, and 25 mM HEPES) in 5 mL polypropylene tubes. Single cells were sorted into 96-well plates filled with 200 μL of D10 medium by FACSAria (BD Biosciences, Franklin Lakes, NJ, USA). Cells were cultured for 14 days and passaged to 24-well plates, 6-well plates, and 10 cm dishes, in this order.

### 2.9. Western Blotting for PLC-β3

Cells or crushed frozen tissues from mice were lysed with 1% NP-40 buffer (20 mM Tris-HCl [pH 8.0], 0.15 M NaCl, 1 mM EDTA) mixed with protease and phosphatase inhibitors (sodium vanadate, phenylmethylsulfonyl fluoride, aprotinin, leupeptin, pepstatin, 4-nitrophenyl 4-guanidinobenzoate, sodium fluoride), homogenized by needles and syringes, resolved by 5–14% gradient acrylamide gels, and transferred to PVDF membranes. Membranes were blocked with 5% non-fat milk, cut horizontally according to the molecular weight of the target protein, incubated with individual primary antibodies (anti-PLC-β3 antibodies (Cell Signaling Technology, Danvers, MA, USA; clone D9D6S; 1/1000 dilution); anti-tubulin (1/1000 dilution)) for 16 h at 4 °C on a nutator. Blots were washed 4 times with TBS-T for 5 min with gentle agitation and then incubated with anti-rabbit-HRP (Cell Signaling Technology) or anti-mouse-HRP antibodies for 1 h at 37 °C on a nutator. Blots were washed 4 times with TBS-T for 5 min with gentle agitation and then excessive water was removed by sandwiching with paper towels. The blots were then incubated with ECL reagents for 1 min (30 s with the front side up and 30 s with the back side up). After removing excessive reagents by sandwiching them with paper towels, blots were wrapped with SARAN wrap, placed in a cassette by tapes along with auto-luminescent marker sheets, and then X-ray films were exposed to blots for 20 s and processed. While processing the first film, the second film was exposed for 2 min; 2 s and 5 s exposures were taken depending on the result of 20 s of exposure and 20 min, 40 min, 2 h, or overnight exposures were taken depending on the result of 2 min of exposure. Processed films were scanned and the brightness of the image was adjusted with Photoshop (Adobe, San Jose, CA, USA) or Powerpoint 2013 (Microsoft, Redmond, WA, USA).

### 2.10. Insert–Deletion Confirmation in Plcb3-Deficient CMT-93 Cells

Genomic DNAs were extracted from approximately 4 × 10^6^
*Plcb3-deficient* (clones 3P11, 3P13, and 2P9, all from sgRNA #3 for exon 2) or WT CMT-93 cell pellets by incubation with 20 μL of Short Genotyping Buffer (2 mM NaCl, 10 mM EDTA, and 0.1% SDS in 50 mM Tris pH 8.0) mixed with 1 μL of Proteinase K (20 mg/mL) for 1 h at 55 °C. Samples were boiled at 95 °C for 5 min to inactivate Proteinase K, and 980 μL of dH_2_O was added to dilute samples. PCR reactions (Indel detection primers are shown below) were performed and analyzed on agarose gels. The bands with the expected size were extracted with a gel extraction kit (QIAGEN, Hilden, Germany) and ligated into zero-blunt vectors (Invitrogen). Ligation products were transformed into *Stbl3*, and plasmids from cultured bacteria were purified by mini-prep (Zymo Research). Plasmid DNAs were sequenced by GENEWIZ.

*Plcb3* Indel detection and sequencing: F: 5′-GAACTGAACCTGCAATGCCC-3′, R: 5′-TCCTTAATTGCCCCACCCAG-3′.

### 2.11. Antibodies

The following antibodies were used for Western blotting. Wnt4 (ab91226, Abcam, Cambridge, UK), Lgr5 (ab75780, Abcam), LRP6 (C47E12, Cell Signaling Technology), Non-phospho (Active) β-catenin (D2U8Y, Cell Signaling Technology), HNF1α (D7Z2Q, Cell Signaling Technology), NFAT1 (D43B1, Cell Signaling Technology), Olfm4 (#66479 or D6Y5A, Cell Signaling Technology), Axin2 (#6163, Prosci; ab109307, Abcam), Actin (H-196, Santa Cruz Biotechnology, Dallas, TX, USA), Dvl2 (30D2, Cell Signaling Technology), Myc-Tag (71D10, Cell Signaling Technology), Phopsho-LRP6 (Ser1490) (#2568, Cell Signaling Technology), PLC-β3 (D9D6S, Cell Signaling Technology), Tubulin (DM1A, Santa Cruz Biotechnology), LRP5 (D80F2, Cell Signaling Technology), Fzd2 (MAB1307, R&D Systems, Minneapolis, MN, USA), Fzd4 (MAB194, R&D), Fzd5 (ab75234, Abcam), Fzd7 (ab64636, Abcam), GPR177 (Wls) (PA5-42570, Invitrogen, Waltham, MA, USA), PORCN (PA5-43423, Invitrogen), Axin (C76H11, Cell Signaling Technology), β-catenin (D10A8, Cell Signaling Technology), CK1 (#2655, Cell Signaling Technology), GSK-3β (27C10, Cell Signaling Technology), FRAT1 (ab108405, Abcam), LGR4 (C-12, Santa Cruz Biotechnology), HA-Tag (C29F4, Cell Signaling Technology), LEF1 (C12A5, Cell Signaling Technology), Anti-rabbit IgG HRP-linked (NA934, GE Healthcare Life Sciences, Chicago, IL, USA), Anti-mouse IgG HRP-linked (NA931, GE Healthcare Life Sciences), and Anti-rat IgG HRP-linked (#7077, Cell Signaling Technology).

### 2.12. RNA Extraction from Plcb3-Deficient CMT-93 and Microarray Data Analysis

RNAs were extracted from subconfluent CMT-93 cultured in a 10 cm dish with RNeasy (QIAGEN) an on-column DNase kit (QIAGEN) following the manufacturer’s instructions. RNA integrity was assessed by 0.8% agarose gel electrophoresis. The gene expression profiles were assessed using a microarray technology with Agilent Sure Print G3 Mouse GE 8 × 60 K Ver 3.0, according to the manufacturer’s instructions (Agilent, Santa Clara, CA, USA. The gMedian fluorescence of probes from individual samples was normalized by R software. Data were analyzed by IPA (QIAGEN Inc., https://www.qiagenbioinformatics.com/products/ingenuity-pathway-analysis). Canonical pathway analysis was performed with a threshold of the activation Z-score > 1.7. A heatmap was created with R software.

### 2.13. qRT-PCR Analysis

cDNA was synthesized from 2.2 μg of RNA with Superscript II (Invitrogen, Cat# 18064071) and Random Hexamer (Invitrogen, Cat# N8080127) following the manufacturer’s protocol. Then, PCR was performed on these cDNAs using Light Cycler (Bio-Rad, Hercules, CA, USA) and following primers. Values calculated by ΔΔCT methods were normalized against housekeeping genes (HPRT or L32). The following validated primers were used.

*Tjp1* (ZO-1): F: 5′-CCTGTGAAGCGTCACTGTGT-3′, R: 5′-CGCGGAGAGAGACAAGATGT-3′; *Cldn1*: F: 5′- TCTACGAGGGACTGTGGATG-3′, R: 5′- TCAGATTCAGCAAGGAGTCG-3′; *Cldn3*: F: 5′-AAGCCGAATGGACAAAGAA-3′, R: 5′-CTGGCAAGTAGCTGCAGTG-3′; *Cldn4*: F: 5′-CGCTACTCTTGCCATTACG-3′, R: 5′-ACTCAGCACACCATGACTTG-3′; *Cldn7*: F: 5′-GCTAAGAAGCCCAACACCAG-3′, R: 5′-TGCAAAATGTACGACTCGGT-3′; *Ocln*: F: 5′-CATAGTCAGATGGGGGTGGA-3′, R: 5′-ATTTATGATGAACAGCCCCC-3′; *Rpl32* (L32): F: 5′-GAAACTGGCGGAAACCCA-3′, R: 5′-GGATCTGGCCCTTGAACCTT-3′; *Wnt2*: F: 5′-ATCTCTTCAGCTGGCGTTGT-3′, R: 5′-AGCCAGCATGTCCTCAGAGT-3′; *Wnt2b*: F: 5′-CACCCGGACTGATCTTGTCT-3′, R: 5′-TGTTTCTGCACTCCTTGCAC-3′; *Wnt4*: F: 5′-AACGGAACCTTGAGGTGATG-3′, R: 5′-GGACGTCCACAAAGGACTGT-3′; *Wnt5a*: F: 5′-CACGCTATACCAACTCCTCTGC-3′, R: 5′-AATATTCCAATGGGCTTCTTCATGGC-3′; *Wls*: F: 5′-CAAATCGTTGCCTTTC-3′, R: 5′-TTGTCACACTTGTTAGGTCCC-3′; *Porcn*: F: 5′-TCCTACATGGCTTCAGTTTCCA-3′, R: 5′-GCGCTTCCGGAGGACAT-3′; *Rspo1*: F: 5′-GCAACCCCGACATGAACAAAT-3′, R: 5′-GGTGCTGTTAGCGGCTGTAG-3′; *Rspo2*: F: 5′-CCAAGGCAACCGATGGAGAC-3′, R: 5′-TCGGCTGCAACCATTGTCC-3′; *Rspo3*: F: 5′-ATGCACTTGCGACTGATTTCT-3′, R: 5′-GCAGCCTTGACTGACATTAGGAT-3′; *Fzd4*: F: 5′-GACAACTTTCACGCCGCTCATC-3′, R: 5′-CCAGGCAAACCCAAATTCTCTCAG-3′; *Fzd7*: F: 5′-ATATCGCCTACAACCAGACCATCC-3′, R: 5′-AAGGAACGGCACGGAGGAATG-3′; *Lgr5*: F: 5′-CCTACTCGAAGACTTACCCAGT-3′, R: 5′-GCATTGGGGTGAATGATAGCA-3′; *Axin2*: F: 5′-GAGTTATCCAGCGACGCACT-3′, R: 5′-CATGCGGTAAGGAGGGACTC-3′; *Olfm4*: F: 5′-GCCACTTTCCAATTTCAC-3′, R: 5′-GAGCCTCTTCTCATACAC-3′; *Ascl2*: F: 5′-CTACTCGTCGGAGGAAAG-3′, R: 5′-ACTAGACAGCATGGGTAAG-3′; *Gja1*: F: 5′-GATCGCGTGAAGGGAAGAAG-3′, R: 5′-CAGCCATTGAAGTAAGCATATTTTG-3′; *Hnf1a*: F: 5′-AGAGCCCCTTCATGGCAACC-3′, R: 5′-TGAAGACCTGCTTGGTGGGTG-3′; *Nfat1*: F: 5′-GTGCAGCTCCACGGCTACAT-3′, R: 5′-GCGGCTTAAGGATCCTCTCA-3′; and *Hprt*: F: 5′-AAGCTTGCTGGTGAAAAGGA-3′, R: 5′-TTGCGCTCATCTTAGGCTTT-3′

### 2.14. Gene Set Enrichment Analysis (GSEA)

The β-catenin-regulated gene set and Lgr4/5 signature gene set were obtained from the previous publications [20,21,22,23,24], and the original KEGG WNT signaling pathway was used. GSEA was performed as described [25,26].

### 2.15. Reanalysis of Publicly Available Single-Cell RNA-Seq Data

Single-cell RNA-seq data for human terminal ileum of childhood-onset Crohn’s disease and matched healthy controls [27] were obtained from https://www.gutcellatlas.org/ as an H5AD file. The data were analyzed using Scanpy 1.8.1 [28]. Mouse intestine datasets [29,30] were analyzed at https://portals.broadinstitute.org/single_cell/study/fasi-immune-mouse-small-intestine and https://singlecell.broadinstitute.org/single_cell/study/SCP44/small-intestinal-epithelium.

### 2.16. ATAC-Seq Analysis

Subconfluent *Plcb3-deficient* (clones 3P11, 3P13, and 2P9) or control (clones Scr1, Scr2, and Scr4) CMT-93 cells were dissociated with Trypsin/EDTA (in 1x PBS pH 7.4) and the numbers of cells were counted. Cells were resuspended with PBS at 1 × 10^6^/mL and 5 × 10^4^ cells were aliquoted and centrifuged at 2000 rpm (500 rcf) for 5 min. The supernatant was removed and cells were resuspended with 50 μL of cold lysis buffer (10 mM Tris-Hcl, pH 7.4, 10 mM NaCl, 3 mM MgCl_2_, 0.1% NP40) [31] in 1.5 mL Eppendorf tubes. Tubes were vortexed for 1 min and centrifuged at 2000 rpm (500 rcf) for 10 min at 4 °C. The supernatant was carefully removed and pellets were resuspended with 20 μL of Tagment DNA (TD) buffer (Illumina, San Diego, CA, USA), and 3 μL of QXT transposase was added to the samples on ice. Samples were incubated at 37 °C for 60 min and then placed on ice for 1 min to cool. A total of 2 μL of 1% SDS (final concentration 0.1%) was added and incubated at RT for 2 min to stop reactions. Then, 30 μL of AMpure beads (Thermo Fisher Scientific, Cat # NC9959336) was added to each sample, which was pipetted 10 times to mix and then incubated for 5 min at RT. Samples were placed on the magnetic stand for 2 min and then the supernatant was removed. Samples were washed with 200 μL of freshly prepared 80% ethanol twice and then air-dried for 10 min after complete removal of the residual ethanol. Pellets were resuspended with 11 μL of nuclease-free water and incubated for 2 min. Samples were placed on the magnet and incubated for 2 min. Then, 10 μL of the samples were transferred to new tubes. A total of 10 μL of the samples was mixed with Kapa-Hifi 2× master mix 25 μL, H_2_O 13 μL, QXT custom 5XX primer 1 μL, and QXT custom 7XX primer 1 μL. A PCR reaction was conducted with the following protocol: 1. 68 °C for 2 min, 2. 98 °C for 2 min, 3. 98 °C for 30 s, 4. 57 °C for 30 s, 5. 72 °C for 1 min, 6. repeat steps 3 to 5 (12 cycles in total), 7. 72 °C 5 min, 8. 4 °C hold. Then, 20 μL of AMpure beads was added to the samples, which were incubated for 5 min and placed on a magnet; then, the supernatants were transferred to other tubes. A total of 30 μL of AMpure beads was added to the samples, which were incubated for 5 min, placed on a magnet, washed twice, and eluted with 13 μL of nuclease-free water. The quality of the library was checked by Tapestation HS 1000 (Agilent) and the amount of usable library was further quantified by qPCR. Samples were pooled for sequencing with HiSeq (Illumina). Sequencing data generated were uploaded to the Galaxy web platform, and the public server at usegalaxy.org was used to analyze the data. Paired-end FASTQ reads were mapped to the mouse genome (mm10) by Bowtie2 [32] with the default setting of Galaxy and then triplicated libraries were merged. Then, files were converted to bigwig files by bamtobigwig. The peaks were named with MACS2 [33]. The mapped reads were visualized with an integrative genome viewer (Broad Institute, Cambridge, MA, USA).

### 2.17. Co-Immunoprecipitation

A total of 25% confluent 293T cells (about 3 × 10^6^ cells) cultured in DMEM (Gibco) supplemented with 10% FBS, penicillin/streptomycin, and glutamine was transfected with a vector control or untagged, N-terminally myc-tagged, or C-terminally myc-tagged human PLC-β3 expression vectors together with expression vectors for a test protein. Plasmids for test proteins including those coding for Tcf4, Lef1, β-catenin, GSK-3β, Axin, and NLK were provided by Dr. Tohru Ishitani. Plasmids for NFAT1 were gifts from Dr. Anjana Rao. Plasmids expressing Gαq were gifts from Dr. Shinya Tsukiji. Constitutively active Gαq mutant (Q209L) was generated using Gibson assembly (NEB) after PCR amplification with primers F: 5′-GGGGCCTGAGGTCAGAGAGAAGAAAATGGA-3′ and R: 5′-CTGACCTCAGGCCCCCTACATCGAC-3′. cDNAs for LGR4 (BC033039), LGR5 (BC096324), LRP6 (BC117136), WLS (BC110826), Dvl2 (BC014844), Frat1 (BC034476), Fzd5 (BC117723), and LRP5 (BC011374) were obtained through transOMIC technologies (Huntsville, AL, USA).

Then, 48 h after transfection, cells were washed twice with PBS and lysed with 1% NP-40 buffer mixed with protease and phosphatase inhibitors. A total of 10 μL of Dynabeads (Thermo Fisher Scientific, Cat. #10003D) was used to preincubate with 1 μg of control mouse IgG1 (MG1-45, BioLegend, San Diego, CA, USA), anti-c-Myc (MC045, Nacalai Tesque, Kyoto, Japan) anti-HA (HA124, Nacalai Tesque), anti-Flag (M2, Sigma-Aldrich), control rabbit IgG (DA1E, Cell Signaling Technology), anti-Dvl2 (30D2, Cell Signaling Technology), anti-TCF4/TCF7L2 (C48H11, Cell Signaling Technology), anti-NFAT1 (D43B1, Cell Signaling Technology), or anti-GSK-3β (27C10, Cell Signaling Technology) antibodies for 30 min at RT and then washed with lysis buffer. Beads were incubated with 100 μg of lysate in 200 μL for 10 min at RT and then washed with 200 μL of lysis buffer 3 times. Beads were resuspended with 40 μL of 1x SDS sample buffer and boiled at 95 °C for 5 min. Co-immunoprecipitated proteins were assessed by Western blotting with the following antibodies: LRP6 (C47E12, Cell Signaling Technology, 1/1000, Lot:3), LRP5 (D80F2, Cell Signaling Technology, 1:1000, Lot:2), Lgr5 (ab75780, Abcam, 1:1000, Lot:GR131564-18), Fzd5 (ab75234, Abcam, 1:1000, GR62870-20), Dvl2 (30D2, Cell Signaling Technology, 1:1000, Lot:4), β-catenin (D10A8, Cell Signaling Technology, 1:1000, Lot:5), Axin (C76H11, Cell Signaling Technology, 1:1000, Lot:4), CK1 (#2655, Cell Signaling Technology, 1:1000, Lot:2), GSK-3β (27C10, Cell Signaling Technology, 1:1000, Lot:14), Frat1 (ab108405, Abcam, 1:1000, Lot:GR49498-3), NFAT1 (D43B1, Cell Signaling Technology, 1:1000, Lot:4), GPR177 (Wls) (PA5-42570, Invitrogen, 1:1000, Lot:PK2296133), NLK (D9X3C, Cell Signaling Technology, 1:1000), Lef1 (C12A5, Cell Signaling Technology, 1:1000, Lot:5), TCF4/TCF7L2 (C48H11, Cell Signaling Technology), 1D4-tag (sc-57432, Santa Cruz Biotechnology, 1:500), and Pan-Gα (3992S, Cell Signaling Technology, 1:1000).

The presence of immunoprecipitated proteins was confirmed by re-probing of the blots after mild stripping with the following antibodies: Myc-Tag (71D10, Cell Signaling Technology, 1:1000, Lot:5), Dvl2 (30D2, Cell Signaling Technology, 1:1000, Lot:4), and TCF4/TCF7L2 (C48H11, Cell Signaling Technology).

### 2.18. 16S rRNA DNA Sequencing

Fecal pellets were collected from mice before and after DSS treatment. DNA was extracted from approximately 20 mg of feces with the QIAamp DNA Stool Mini Kit (QIAGEN), following the manufacturer’s protocol. Libraries were prepared following a 16S metagenomic sequencing library preparation guide (Illunina) and run with Miseq v3 (Illumina). Generated FASTQ reads were processed with Illumina 16S metagenomics 1.0.1 pipeline (Illumina; basespace application). The reads classified to the individual genus and species were divided by the total number of classified reads to calculate the proportion.

### 2.19. Animal Experiments

FITC injection experiment. Mice were treated with water or 3% DSS ad libitum for 24 h and orally gavaged with FITC-Dextran (70 kDa). Then, 6 h later, blood was collected from the mesenteric vein and the fluorescence of FITC-Dextran in sera was measured by SpectramaxM2.

*Citrobacter* infection experiment. Next, 1 × 10^9^ CFU of chloramphenicol-resistant *Citrobacter rodentium* [34] was orally injected and tissues were harvested 6 days later. The burden of bacteria in liver, spleen, and fecal samples from the colons was assessed with chloramphenicol plates by inoculating serially diluted tissue homogenates.

Antibiotics treatment. Mice were treated with a cocktail of antibiotics (ampicillin 1 mg/mL, neomycin 1 mg/mL, vancomycin 0.5 mg/mL, and metronidazole 1 mg/mL) with a sweetener (Splenda; 10 mg (*w*/*v*)) for 4 weeks before DSS treatment.

Human R-spondin1 treatment. Mice were intravenously injected with PBS or 100 μg or 200 μg of human R-spondin1 (a gift from Kyowa-Kirin, Tokyo, Japan) in 200 μL of PBS every 24 h. For Western blotting and qRT-PCR, tissues were harvested 6 h after the last i.v. injection.

Bone marrow chimera experiment. Next, 1.0 × 10^6^ cells were collected from donor WT or *Plcb3^−/−^* mice and intravenously injected into X-ray-irradiated (480 rad × 2; 960 rad in total) recipient WT or *Plcb3^−/−^* mice. Chimeric bone marrow was confirmed by blood genotyping and mice were orally treated with 3% DSS 9 weeks after the bone marrow transfer.

Histological analysis. Mouse intestines from the stomach to the colon were harvested at days 2, 4, or 7 of DSS treatment. After measuring the length of the intestines and taking photos, the intestines were cut open longitudinally, washed, rolled up, and fixed with 4% zinc formalin for 24 h. Then, formalin was replaced with 70% isopropanol (*v*/*v*) and tissues were kept in the solution for at least 24 h. The samples were further dehydrated with 70% ethanol, 80% ethanol, 90% ethanol, 100% ethanol, and Pro-Per; then, the samples were embedded in paraffin and sectioned with microtome at a 5 μm thickness. Slides were stained with H&E, PAS, or Alcian blue and scanned by Axioscan Z1 (ZEISS). Histological scores were calculated in a blinded manner as previously described [35]. Colons were separated into 3 areas and the extent of inflammation and crypt damage was assessed for each area.

Whole mount staining of ileum. A total of 10 cm of ileum harvested from water or DSS-treated mice was washed and fixed with 4% paraformaldehyde (PFA) for 1 h, washed, and placed in Magic buffer (2% FBS, 0.5% Saponin, 0.1% Sodium Azide in PBS). Muscles were carefully scraped with forceps under a stereoscope and tissues were cut into 2 mm × 2 mm pieces; permeabilized with Magic buffer for 2 h; incubated with anti-Olfm4 (#66479, Cell signaling Technology) or Ki-67 (D3B5, Cell Signaling Technology) for 16 h at RT; incubated with Hoechst 33342 (H3570, Invitrogen), anti-rabbit IgG DyLight^TM^ 488 (#406404, BioLegend), and Alexa Fluor™ 647 Phalloidin (A22287, Invitrogen) for 2 h at RT; and mounted on the slides with Prolong gold. For UEA-1-FITC staining or the analysis of *Lgr5-EGFP-IRES-CreERT2^+^* mice, the incubation step with the primary antibody was skipped. Images were taken by SP5 (Zeiss, Jena, Germany), Fluoview 10i (Olympus, Tokyo, Japan), or LSM880 airyscan (Zeiss) and processed with the Leica Application Suite (for SP5) or Bitplane-Imaris (SciTech, Preston, Australia) for 3D reconstruction. Videos of 3D reconstructed images were generated and exported through Imaris and saved in MP4 format. The numbers of bifurcated crypt lumen and non-bifurcated crypt lumen were counted.

Frozen-/paraffin-embedded-section immuno-fluorescent staining. A total of 10 cm of jejunum or ileum was harvested, washed with 4% PFA by flashing, cut open longitudinally, and fixed with 4% PFA for 1.5 h on ice. Tissues were washed with PBS 3 times and incubated with 15% or 30% sucrose–PBS for 15 or 45 min, respectively, for cryoprotection. Tissues were embedded in OCT compound (Sakura Finetek, USA, Torrance, CA, USA) and sectioned at 10 μm thicknesses. Slides were washed with PBS; permeabilized with 0.3% Triton X for 10 min; blocked with 10% normal donkey serum for 1 h; incubated with anti-ZO-1 (H-300, Santa Cruz Biotechnology), claudin4 (ZMD.306, Invitrogen), and anti-lysozyme (C-19, Santa Cruz Biotechnology) for 16 h at 4 °C; incubated with secondary antibodies for 1 h at RT; and mounted with Prolong gold (Fisher Scientific Cat. # P36390). Images were taken with FluoView FV10i (Olympus), Nikon 80i, or Axioscan Z1 and processed with Zen blue (Zeiss). For both samples and corresponding controls, the same microscopy with the same laser intensity was used for each experiment and images were processed in the same way throughout the study. For paraffin-embedded human samples, antigen retrieval was performed by heating for 10 min at 95 °C in 10 mM sodium citrate solution (pH 6.0) before permeabilization. Anti-PLC-β3 (D9D6S, Cell Signaling Technology) antibody or a rabbit IgG isotype control (DA1E, Cell Signaling Technology) was used as the primary antibody. Axioscan Z1 was used for initial quantitative analysis and FluoView FV10i was used for conformation and taking representative images.

### 2.20. Drosophila Immunohistochemistry

Third instar larval wing imaginal discs were dissected in PBS and fixed in 4% paraformaldehyde in PBS for 20 min. After fixation, discs were washed with PBS with 0.1% Triton X-100, followed by incubation in PBS with 0.1% Triton X-100 and 10% BSA for 1 h at RT. Wing discs were incubated with rabbit anti-dsRed (Clontech/TaKaRa, Cat. no. 632496, 1:500) to detect the *senseless* transcriptional reporter *mScar:T2A:sens* [36], mouse anti-Wg (4D4, Developmental Studies Hybridoma bank [DSHB], 1:500), and mouse anti-Engrailed (En) (4D9, DSHB, 1:20) at 4 °C overnight in PBS with 0.5% Triton X-100, followed by incubation with secondary antibodies (anti-mouse and anti-rabbit Alexa Fluor 488 and 555 conjugates; Thermo Fisher Scientific, 1:500) for 2 h at RT. Specimens were stained with DAPI (2 μg/mL) and mounted in Prolong Gold (Invitrogen). Images were captured using a Nikon CSU-W1 spinning disk confocal microscope and processed with Adobe Photoshop/Illustrator software. Quantitation was performed using Prism 8 software (Graphpad, Boston, MA, USA).

### 2.21. Drosophila Abdomens

Adult abdomens from crosses between the *B119-Gal4* driver and denoted RNAi lines were dissected and mounted onto glass slides and covered with glass coverslips. Images were acquired using a Leica MZ10F modular stereo microscope and a Zeiss Axiocam 208 camera. Images were processed with Zen Blue 3.0 and Adobe Photoshop/Illustrator software. Quantitation was performed using Prism 8 software (Graphpad).

### 2.22. Statistical Analysis

No statistical test was performed to pre-determine sample sizes. A log-rank test, two-tailed Student’s *t*-test, paired *t*-test, Mann–Whitney U test, one-way ANOVA, or Kruskal–Wallis and Dunn’s multiple comparison tests were performed through Prism (Graphpad) where applicable. *p* values smaller than 0.05 were considered statistically significant.

## 3. Results

### 3.1. PLC-β3 KO Mice Exhibit DSS-Induced Lethality with Severe Inflammation in the Small Intestine and Anemia

All *Plcb3^−/−^* mice exposed to 3% DSS in drinking water lost body weight (Appendix A) and died within 6 days, while WT mice survived (Figure 1A). DSS-treated *Plcb3^−/−^* mice exhibited bloody stool (Figure 1B) and severe anemia (Figure 1C). *Plcb3^−/−^* mice exhibited increased intestinal permeability with or without DSS exposure, as demonstrated by FITC-dextran (70 kDa) gavage (Figure 1D) and *Citrobacter rodentium* infection (Appendix A). These findings were consistent with DSS-induced morphological abnormalities including rough and disrupted villus tips of the jejunum and ileum (Figure 1E) and reduced expression and/or aberrant distribution of several tight junction proteins including ZO-1, occludin, and claudins 1/3/4/7, even without DSS treatment (Figure 1F,G). Similar to their expression in the IECs, crypt organoids from the small intestine of *Plcb3^−/−^* mice exhibited decreased mRNA expression of several tight junction proteins including ZO-1, occludin, and claudins 1/3/4/5/8/9/10/11 (Appendix A). Some of these and other proteins were confirmed by Western blotting, immunofluorescence confocal microscopy, and qRT-PCR (Appendix A). Therefore, the increased intestinal permeability in *Plcb3^−/−^* mice was likely due to IEC-intrinsic impairment. The differences in DSS-induced injury between WT and *Plcb3^−/−^* mice were limited to the small intestine, as the injury scores in the colons of DSS-exposed *Plcb3^−/−^* mice were comparable to those of WT mice (Appendix A). No abnormalities were observed in heterozygous *Plcb3^+/−^* mice.

Analysis of IECs by fluorescence microscopy and 3D reconstitution revealed that the numbers of Olfm4^+^ crypt base columnar cells, which were Lgr5^+^ ISCs [37] (Figure 1H and Appendix A), and Ki-67^+^ proliferative cells (Figure 1I and Appendix A) were significantly reduced in the ilea of untreated and DSS-treated *Plcb3^−/−^* mice. Reduced numbers of Lgr5^+^ ISCs were also observed in DSS-treated *Plcb3^−/−^;Lgr5-EGFP-IRES-CreERT2* reporter mice compared with control *Plcb3^+/−^;Lgr5-EGFP-IRES-CreERT2* mice (Appendix A). Consistent with the reduction in proliferative cells, crypt fissions were drastically reduced in *Plcb3^−/−^* mice (Appendix A). In addition, increases in crypt goblet cells (Figure 1J and Appendix A) and mucin (Appendix A) were observed. Furthermore, reduced expression of *lyz1*, *Reg3a*, *Reg3g*, and lysozyme protein (Appendix A) and reduced α(1,2)fucosylation (Figure 1K and Appendix A) indicated abnormalities in Paneth cells [38].

### 3.2. Multiple Non-Hematopoietic Cell Types Contribute to the DSS-Induced Lethality of Plcb3^−/−^ Mice

Single-cell RNA sequencing data of human [27,39] and mouse [29] intestinal cells indicate that *PLCB3/Plcb3* mRNA is expressed most highly in IECs, other non-hematopoietic cells including fibroblasts and endothelial cells, and hematopoietic cells (Appendix A). Furthermore, *PLCB3/Plcb3* expression is high in the early stages of epithelial cell development and in enterocytes, whereas enteroendocrine cells and goblet cells tend to lose this expression (Appendix A). Thus, abnormalities in IECs likely contribute to the DSS-induced lethality in *Plcb3^−/−^* mice. The involvement of IECs and other cell types in this phenotype was tested initially with bone marrow chimeras. Irrespective of the donor genotype, the recipient *Plcb3^−/−^* mice, but not the recipient WT mice, died upon DSS exposure (Figure 2A). This result suggests that non-hematopoietic cells are responsible for the lethal phenotype of *Plcb3^−/−^* mice. To further identify the cell type(s) responsible for the DSS-induced lethality, we made *Plcb3^fl/fl^* mice to generate conditional knock-out (cKO) mice (Appendix A). IEC-specific deletion of the *Plcb3* gene was achieved by crossing *Villin-Cre* mice with *Plcb3^fl/fl^* mice (termed *Plcb3^ΔIEC^* mice, Appendix A). *Plcb3^ΔIEC^* mice exhibited severe anemia when fed 5% DSS (Appendix A) but not when fed 3% DSS (data not shown) and slightly increased intestinal permeability (Appendix A). In comparison to *Plcb3^fl/fl^* mice, *Plc3^ΔIEC^* mice were more susceptible to DSS when pretreated with antibiotics (Appendix A), but did not display the early lethal phenotype without antibiotic treatment (Figure 2B). Furthermore, we generated fibroblast (FB)- (using Col1a2-CreERT2), endothelial cell (EC)- (using Cdh5-Cre), myeloid cell (MC)- (using LysM-Cre), dendritic cell (DC)- (using Cd11c-Cre), and type 3 innate lymphoid cell (ILC3)- (using Rorc-Cre) specific cKO mice. As expected from our results with bone marrow chimeras, the resistance of MC-, DC-, and ILC3-specific cKO mice to DSS was similar to that of WT mice (Figure 2C). No single cKO mouse showed the early lethal phenotype (Figure 2C). Interestingly, *Plcb3^ΔFB^* mice exhibited goblet cell hyperplasia (Appendix A), similar to *Plcb3^−/−^* mice. We then intercrossed single cKO strains to make double cKO strains and found that only *Plcb3^ΔIEC^^;EC^* showed early intestinal bleeding following treatment with 3% DSS (Appendix A). Finally, we crossed *Plcb3^ΔFB^* and *Plcb3^ΔIEC^^;EC^* to generate triple cKO mice, which exhibited increased DSS susceptibility (Figure 2D,E). Thus, these results suggest that PLC-β3-regulated signaling pathways in IECs, FBs, and ECs work together to protect from DSS-induced injury and/or promote repair.

### 3.3. Small Intestinal Epithelial Cells from Plcb3^−/−^ Mice Exhibit Decreased Wnt/β-Catenin Signaling

To gain insight into the molecular mechanisms by which *Plcb3* deficiency increases DSS susceptibility, gene expression in isolated small intestinal IECs was analyzed using DNA microarray. Consistent with reduced proliferative IECs (Figure 1I), gene set enrichment analysis (GSEA) showed that cell cycle-related E2F target genes, G2/M checkpoint genes, and Myc-regulated genes were downregulated in IECs of *Plcb3^−/−^* and *Plcb3^ΔIEC^* mice vs. their respective controls under homeostatic conditions (Appendix A). Importantly, GSEA also revealed reduced expression of Wnt/β-catenin signature genes [20] including *Wls*, *Fzd7*, *Axin2*, *Lgr5*, and *Olfm4* (Figure 3A,B) and the Lgr4-, Lgr4^+^/Lgr5^+^-, and Ascl2^+^-stem/progenitor signature genes [21] in small intestinal IECs of *Plcb3^−/−^* mice under homeostatic (Figure 3A). Similarly, small intestinal IECs from *Plcb3^ΔIEC^* mice under homeostatic conditions exhibited reduced expression of Wnt/β-catenin signature genes and Lgr4^+^-, Lgr4^+^/Lgr5^+^-, and Ascl2^+^-stem/progenitor signature genes in comparison to *Plcb3^fl/fl^* mice (Figure 3C). Lgr4 and Lgr5 bind to R-spondins, which bridge and internalize the negative regulators Rnf43 and Znrf3 with Lgr4/Lgr5 to promote the activation of Wnt/β-catenin signaling [40]. Lgr4 hypomorphic mice exhibit a lethal inflammatory phenotype with reduced ISCs and Paneth cells upon DSS exposure [41], similar to *Plcb3^−/−^* mice. Moreover, Ingenuity Pathway Analysis (IPA) [42] indicated the downregulation of Wnt/β-catenin signaling in *Plcb3^−/−^* mice (Appendix A) and the opposite regulation of β-catenin-regulated genes (Appendix A). Consistent with the involvement of mesenchymal cells in the DSS-induced lethality of *Plcb3^−/−^* mice, small intestinal mesenchymal cells in *Plcb3^−/−^* mice exhibited reduced mRNA expression of several Wnt ligands, as well as factors essential for Wnt secretion (Wls, Porcn) and Wnt signal enhancers (R-spondins 1/2/3) (Figure 3D and Appendix A). Reduced mRNA expression of the Wnt target genes *Wls*, *Fzd7*, *Axin2*, *Lgr5*, *Olfm4*, and *Ascl2* in *Plcb3^−/−^* IECs was found, several of which were confirmed by qRT-PCR (Figure 3B). Furthermore, reduced levels of Wls, Wnt-4, LRP6, Fzd2, Axin2, β-catenin, GATA6, and Olfm4 proteins in *Plcb3^−/−^* IECs were confirmed by Western blotting (Figure 3E). By contrast, SPDEF, which antagonizes the expression of Wnt/β-catenin target genes [43], was increased in *Plcb3^−/−^* IECs. These results collectively show that PLC-β3 regulates expression of the Wnt/β-catenin signaling pathway at numerous levels, including the Wnt ligands, regulators of Wnt secretion (Wls, Porcn), Wnt signal enhancers (R-spondin 1/3), Wnt co-receptors (Fzd2, Fzd7, Lgr5, LRP6), the negative feedback regulator (Axin2), and transcription factors (Ascl2, GATA6, SPDEF).

### 3.4. PLC-β3 Regulates the Wnt/β-Catenin Signaling Pathway in IECs at Multiple Regulatory Levels

To further explore the function of PLC-β3 in IECs, *Plcb3*-deficient CMT-93 mouse IEC lines were generated using the CRISPR/Cas9 method (Figure 4A). GSEA and IPA canonical pathway analysis of transcriptomes in three *Plcb3*-deficient clones with different mutations vs. three *Plcb3*-sufficient controls showed reduced Wnt/β-catenin signaling (Figure 4B,C), which was ranked as one of the most affected among the *Plcb3*-affected pathways (Appendix A). The reduced expression of several Wnt/β-catenin signaling molecules was confirmed by qRT-PCR and Western blotting (Figure 4D,E). Consistent with these observations, the transcriptional activity of TCF/LEF was markedly reduced in *Plcb3*-deficient cells (Figure 4F). Moreover, ATAC-seq analysis [44] followed by pathway analysis using clusterProfiler [45] revealed that the Wnt signaling pathway, inflammatory bowel disease pathway, calcium signaling pathway, and tight junction pathway were in the top 30 pathways that were different between WT and *Plcb3^−/−^* cells (Appendix A). Among them, Wnt/β-catenin pathway gene loci, including *Dkk2*, *Fzd7*, *Axin2*, and *Lgr5*, were identified (Figure 4G). Additionally, numerous other loci near *Gja1* (*Cx43*), *Ttc9*, *Arhgap26*, and *Robo1* were identified as chromatin accessibility sites regulated by PLC-β3 (Appendix A). Gja1 plays a role in the innate immune control of commensal-mediated wound repair of IECs [46], although enteric glial Gja1 does not affect acute DSS-induced colitis [47]. Robo1, the receptor for the secreted glycoprotein Slit2, mediates DSS-induced colitis by activating the autophagy of Lgr5^+^ ISCs [48]. The above results collectively suggest that the expression of numerous Wnt/β-catenin pathway genes is controlled by PLC-β3 in IECs at the transcriptional and epigenetic levels.

PLC-β3, composed of several protein-interacting domains (Appendix A), directly interacts with several proteins, such as GTP-bound Gαq subunit [49], and is activated by GPCRs. Some Fzd proteins act as Gαq-coupled receptors, similar to prototypical GPCRs, raising the possibility that an Fzd protein(s) and Gαq interact with PLC-β3. Thus, we next examined the possibility that PLC-β3 regulates the Wnt/β-catenin signaling pathway by protein–protein interactions. For this purpose, three methods were employed: first, co-immunoprecipitation experiments were performed using lysates from HEK293T cells expressing untagged or myc-tagged PLC-β3 and Wnt/β-catenin signaling proteins (Figure 4H–J and Appendix A); second, GST fusion pulldown experiments were performed using PLC-β3 domains (Appendix A); and third, proximity ligation assays (PLAs) were applied to HEK293T transfectants (Figure 4K). Co-immunoprecipitation experiments using anti-Myc antibodies revealed that myc-tagged PLC-β3 can interact with several Wnt signaling proteins including Fzd1 (but not Fzd4/5/7/8) (Appendix A), as well as Wls, Dvl2, NLK (which phosphorylates and activates Lef-1, downstream of Dvl2 [50]), GSK3β, Tcf-4, and Lef-1 (Figure 4H). C-myc-tagged PLC-β3 was co-immunoprecipitated with constitutively active Gαq regardless of the presence of different Fzd proteins (Fzd1/4/5/7/8), but only Fzd1 co-immunoprecipitated with WT Gαq. The interaction of Fzd1 with C-myc-tagged PLC-β3 was reduced by constitutively active Gαq (Appendix A), and Fzd1 seemed to be stabilized by PLC-β3 (Appendix A). PLC-β3′s co-immunoprecipitation with anti-Dvl2 was robust (Figure 4I), whereas its co-immunoprecipitation with anti-Tcf-4 antibodies was weak (Figure 4J). Strong co-immunoprecipitation of PLC-β3 with Dvl2, a scaffold protein that interacts with Fzd and LRP5/6 at the plasma membrane [51], suggests that PLC-β3 may participate in the signaling complex composed of these proteins. Dvl2 and GSK3β were pulled down with PH-EF and split catalytic domains of PLC-β3. Fzd1 was pulled down with the C-terminal portion of PLC-β3, while β-catenin and Lgr5 were pulled down with the PH-EF domain of PLC-β3 (Appendix A). The interaction between PLC-β3 and Lef-1 was confirmed by PLA (Figure 4K). These results raise the possibility that PLC-β3 promotes Wnt/β-catenin signaling by interacting not only with the Wnt receptor complex but also with other Wnt signaling proteins, in addition to the transcriptional and epigenetic regulation of the pathway proteins.

To assess the effect of persistent Wnt/β-catenin signaling on IECs in vivo, mice were intravenously treated with R-spondin 1 and, 24 h later, Wnt/β-catenin signaling molecules were examined. Protein and/or mRNA expression of LRP6, Axin2, β-catenin, Lgr5, Olfm4, and Ascl2 was more robustly increased in the small intestinal IECs of WT mice than in those of *Plcb3^−/−^* mice (Figure 4L,M). Furthermore, the phosphorylation of LRP6 at Ser1490, which is an indicator of the activation of Wnt/β-catenin signaling [52,53,54], was strongly induced in response to R-spondin 1 in WT, but not in *Plcb3^−/−^* mice (Figure 4M). These results indicate that PLC-β3 promotes Wnt/β-catenin signaling in IECs.

### 3.5. Dysbiosis May Also Contribute to DSS Susceptibility in Plcb3^−/−^ Mice

As antibiotic-treated *Plc3^ΔIEC^* mice were susceptible to DSS (Appendix A), we further evaluated the possible effect of intestinal microbiota on DSS susceptibility. DSS treatment of co-housed WT and *Plcb3^−/−^* mice (Figure 5A), littermates from the same *Plcb3^+/−^* breeding pairs (Figure 5B), or antibiotic-treated mice (Figure 5C) indicated that the lethality of *Plcb3^−/−^* mice was largely independent of microbiota. However, microbiota-mediated protection was lost to some degree in *Plcb3^−/−^* mice, as these mice had an increased number of crypt abscesses in the small intestine (Figure 5D,E), and *Plcb3^−/−^* and *Plc3^ΔIEC^* mice exhibited reduced expression of commensal microbe-induced genes compared with their controls (Figure 5F). 16S rRNA sequencing revealed the existence of pathogenic *Escherichia albertii* and *Escherichia coli* in *Plcb3^−/−^* mice (Figure 5G), and a dramatic increase of these pathogenic species was present after DSS treatment (Figure 5H,I). To further address the effect of microbiota on DSS susceptibility, WT mice were fostered by *Plcb3^−/−^* mice at the date of birth. Fostered WT mice showed an increased number of the *Escherichia* species (Figure 5G–I) and earlier body weight loss upon DSS treatment compared to WT mice raised by WT mice (Figure 5J). These results suggest that dysbiosis caused by PLC-β3 deficiency contributes to increased susceptibility to DSS.

### 3.6. Reduced Expression of PLC-β3 and Altered Expression of PLC-β3-Regulated Genes in Ileal Crohn’s Disease

To validate the clinical relevance of PLC-β3 levels in human IBD patients, transcriptome data from published studies were analyzed. In the GSE57945 dataset [55], *PLCB3* mRNA expression in ileal biopsy samples was lower in pediatric CD patients, especially those with deep ulcers, but not in patients with ulcerative colitis (Figure 6A and Appendix A). Data in GSE83448 [56] also showed reduced *PLCB3* mRNA expression in adult ileal CD patients (Appendix A) and data in GSE179285 [57] showed reduced *PLCB3* mRNA more strongly in inflamed than in uninflamed terminal ileum of adult CD, but not in colonic CD or ulcerative colitis patients (Appendix A). Pediatric colon biopsies also showed no reduction in *PLCB3* expression in the GSE126124 dataset [58] (Appendix A). Despite the cellular heterogeneity of biopsy samples, reduced PLC-β3 protein in three out of seven ileal CD patients compared to healthy controls was confirmed by immunofluorescent staining (Figure 6B,C). The expression of ileal CD signature genes [55] was similarly altered in *Plcb3^−/−^* mice (Figure 6D). For example, the expression of *PLCB3* mRNA in CD patients showed an inverse correlation with that of *DUOX2*, an upregulated CD signature gene, and a positive correlation with that of *APOA1*, a lipoprotein downregulated in CD (Appendix A). Conversely, the expression of PLC-β3 signature genes as defined by ≥2-fold changes and false discovery rate (FDR) q-values < 0.01 in *Plcb3^−/−^* vs. WT mice (Appendix A) was similarly altered in CD biopsy specimens, especially those with deep ulcers (Figure 6E). Together with data from *Plcb3*-deficient mice and IECs, these results suggest that the reduced expression of PLC-β3 is causally linked to ileal CD.

### 3.7. PLC-β Promotes Wnt/Wingless Signaling in Drosophila

The Wnt signaling pathway is evolutionally conserved [59]. In *Drosophila melanogaster*, the only PLC-β isozyme is encoded by the gene *Plc21C*. Wingless (a Drosophila Wnt) specifies the fate of cells that form sensory bristles at the adult wing margin, in part through the expression of the Wingless target gene *senseless* (*sens*) [60]. RNAi-mediated knockdown of *Plc21C* in the posterior compartment of the wing disc using the *hedgehog (hh)-Gal4* driver decreased Sens levels solely in the posterior compartment in more than 90% of discs (marked by Engrailed, Figure 7A,B). Multiple independent RNAi constructs that target different regions of *Plc21C* reduced Sens specifically in the posterior compartment, showing a robust effect. By contrast, RNAi-mediated knockdown of a control gene *yellow (y)* resulted in minimal Sens reduction (Figure 7A,B), supporting the specificity of the *Plc21C* RNAi results. Moreover, RNAi knockdown of *Plc21C* did not inhibit *wingless* expression (Figure 7C), indicating that the reduction in Sens was not due to decreased Wingless levels but instead resulted from downstream defects in the Wingless signaling pathway.

Inactivation of core Wingless signaling pathway components, including Wingless, Armadillo/β-catenin, and Armadillo’s transcriptional co-activators dTCF/Pangolin and Legless results in the loss of bristle-bearing cuticular plates known as sternites and sternal bristles in the ventral abdomen [61,62,63]. When two independent RNAi constructs targeting *Plc21C* were expressed using the *B119-Gal4* driver, we observed a marked loss of sternites and sternal bristles, as well as an expansion of pleura (Figure 7D,E). A severe phenotype occurred in at least 70% of the flies analyzed. This reduction in abdominal sternites was rarely observed with RNAi targeting of the control *y* gene (Figure 7D,E). These results suggest that PLC-β-mediated regulation of the Wnt signaling pathway is conserved from *Drosophila* to mice and humans.

## 4. Discussion

Based on mouse IBD models, molecular analyses of IECs, and ileal biopsy specimens from human CD patients as well as extensive transcriptomic and epigenetic datasets, our study suggests that PLC-β3 promotes intestinal homeostasis and repair after injury through the activation of the Wnt/β-catenin signaling pathway. Wnt/β-catenin signaling drives the cell cycle and, conversely, the cell cycle influences Wnt signaling. During the cell cycle, the expression of β-catenin oscillates, and the maximal activation of β-catenin and *Axin2* expression occurs at the G2/M transition [64,65,66]. Moreover, the mitosis-specific Cdk14-cyclin Y kinase complex phosphorylates LRP6 at Ser-1490 during G2/M, which promotes the activation of signaling [64]. Surprisingly, PLC-β3 regulates Wnt/β-catenin signaling through multiple regulatory mechanisms, including transcriptional, epigenetic, and, potentially, protein–protein interaction levels. Such multi-level regulatory mechanisms underlie the important role of PLC-β3 in IECs of the mouse and human small intestine. Furthermore, PLC-β regulates Wnt/Wingless signaling in *Drosophila*. Thus, we conclude that the newly identified PLC-β-mediated regulation of Wnt/β-catenin signaling is evolutionally conserved from *Drosophila* to humans.

As PLC-β3 deficiency leads to reduced β-catenin expression in IECs, PLC-β3 functions upstream of this protein within the Wnt/β-catenin pathway. One of the ten Frizzled receptors and LRP5/LRP6 act as Wnt receptor complexes [67]. Fzd is a seven-transmembrane domain receptor, like conventional GPCRs. Since PLC-β isozymes are linked to GPCRs [10], PLC-β3 likely functions similarly in the Wnt/β-catenin pathway. Consistent with this notion, Fzd1 was co-immunoprecipitated with PLC-β3. Moreover, Dvl2, which works as an oligomerizable scaffold to interact with an Fzd at the plasma membrane [51], was robustly co-immunoprecipitated with PLC-β3. R-spondins bind to their receptors Lgr4/5/6 and bridge the negative regulators Rnf43 and Znrf3 with these Lgr receptors to promote the activation of Wnt/β-catenin signaling [40]. It is known that Wnt-3a mediates the production of phosphatidylinositol 4,5-bisphoshate, the PLC substrate, to regulate LRP6 phosphorylation [68]. All these results along with those of other studies strongly support that PLC-β3 promotes Wnt/β-catenin signaling by interacting not only with the Wnt receptor complex but also with other Wnt signaling proteins.

Downstream of Wnt binding to Fzd, a PLC is thought to generate two messengers: diacylglycerol and inositol 1,4,5-trisphosphate in the Wnt/Ca^2+^ pathway [69]. PLC-β3 may play this role in IECs, as suggested by the IPA of the transcriptomes of PLC-β3-deficient CMT-93 cells (Appendix A). If this is the case, PLC-β3 may be shared by both the Wnt/β-catenin and Wnt/Ca^2+^ signaling pathways.

Our study identified a PLC-β3-dependent Wnt/β-catenin pathway in mouse small intestinal IECs. As PLC-β3 is expressed broadly, it is likely that this pathway functions in other cell types such as fibroblasts and endothelial cells. Changes in the connective tissue are well known in Crohn’s disease, including fat-wrapping due to adipocyte hyperplasia, fibrosis, muscularization, and neuronal and vascular changes [70]. Of note, we showed increased goblet cells in *Plcb3^ΔFB^* mice and DSS-induced early intestinal bleeding in *Plcb3^ΔIEC;EC^* mice, implying the role of fibroblasts and epithelial–endothelial cell interactions in goblet cell differentiation and the blood vessel barrier, respectively. Wnt/β-catenin signaling is implicated in fibrogenesis in a variety of tissues. It was shown that the Wnt/β-catenin signaling pathway is necessary for TGF-β-mediated fibrosis, and the interaction of both pathways is important in the pathogenesis of fibrotic diseases [71]. Wnt signaling stimulated the differentiation of resting fibroblasts into myofibroblasts, increased the release of extracellular matrix components, and induced fibrosis. The expression of Smad6, an inhibitory Smad that negatively regulates signaling downstream of TGF-β, was reduced in the ileum of *Plcb3^−/−^* mice (Appendix A). On the other hand, PLC-β3 works as a negative regulator of VEGF-mediated vascular permeability by regulating intracellular Ca^2+^ release [72]. The ETS transcription factor Erg drives the expression of VE-cadherin and controls junctional integrity in endothelial cells. Birdsey et al. showed that the constitutive endothelial deletion of Erg in mice causes embryonic lethality with vascular defects [73]. Erg controls the Wnt/β-catenin pathway by promoting β-catenin stability. Thus, Erg is an essential regulator of angiogenesis and vascular stability through Wnt signaling. However, Erg expression in the ileum was not significantly affected by PLC-β3 deficiency. Despite these observations showing nonredundant roles of PLC-β3 among different non-hematopoietic cells, triple *Plcb3* cKO mice lacking *Plcb3* in IECs, fibroblasts, and endothelial cells were less susceptible to DSS than *Plcb3^−/−^* mice, suggesting that this pathway also operates in other mesenchymal cells and/or IEC–mesenchymal cell interactions. Results of single-cell RNA sequencing (Appendix A) support this possibility.

Studies of gene expression and chromatin accessibility divided CD into two classes: colon and ileal subtypes [74]. Characteristic features of ileal CD include the reduced expression of α-defensins HD-5 and HD-6, which are expressed by Paneth cells [75], and low Wnt/β-catenin signaling, as exemplified by low expression or genetic variants of TCF-4 [76,77], TCF-1 [78], and LRP6 [79]. Consistent with these previous studies and our current data with *Plcb3^−/−^* mice, we found reduced expression of *PLCB3* and altered expression of *PLCB3*-regulated signature genes in ileal CD, but not in other CD subtypes or ulcerative colitis. This is consistent with our observations that *Plcb3^−/−^* mice exhibited normal histology of the colon under homeostatic conditions and the DSS-induced severe pathology was restricted largely to the small intestine. Therefore, the pathogenic mechanism of ileal CD characterized as a “Paneth cell disease” is likely different from that of colonic CD.

In addition to the reduced Wnt signaling in IECs of ileal CD, macrophages and monocytes are known to play critical roles in IBD pathogenesis [80,81] through the production of inflammatory cytokines such as TNF, IL-1β, IL-6, and IL-23. The upregulation of a set of inflammatory response genes that include these cytokine genes was found in monocytes from IBD patients, and the expression of these genes was inversely correlated with miR-374a-5p expression [82]. Increased expression of miR-374a-5p can suppress monocyte-/macrophage-driven inflammation in all subtypes of IBD. Interestingly, Martin et al. found a pathogenic cellular module termed GIMATS, comprised of inflammatory macrophages, activated DCs, highly activated T cells, IgG^+^ plasma cells, activated fibroblasts, and ACKR1^+^-activated endothelial cells, and a GIMATS^high^ subset of ileal CD patients showed resistance to anti-TNF therapies [39]. Another single-cell analysis of colonic mucosae from pediatric colitis and IBD suggested a defective cAMP response and colonic mucosal platelet aggregates as underlying pathologies, which were ameliorated by dipyridamole, a phosphodiesterase inhibitor, in a pilot clinical trial [83]. These systems biology approaches suggest the involvement of both hematopoietic and non-hematopoietic cells in mucosal inflammation as key IBD pathogenic processes. Although our study showed increased expression of IL-1β and IL-6 mRNAs in the ilea of *Plcb3^−/−^* and *Plcb3^ΔIEC^* mice compared with control mice, the miR-374a-5p-related monocyte’s inflammatory module was not significantly related to the *Plcb3^−/−^* phenotype (data not shown). However, future studies should address the relationship between GIMATS, the monocyte’s inflammatory response module, and PLC-β3-dependent Wnt/β-catenin signaling. Further refinement of our understanding of the pathogenesis of the CD subtypes will help predict responses to therapeutics. Furthermore, it may be worth testing Wnt agonists [8,84] as a potential therapy for ileal CD unless *PLCB3* is totally absent.

The effect of small intestinal microbiota on DSS susceptibility as well as the effect of PLC-β3 deficiency on small intestinal microbiota were rather subtle in *Plcb3^−/−^* mice. By contrast, the protective effect of small intestinal microbiota was robust in *Plcb3^ΔIEC^* mice compared with *Plcb3^fl/fl^* mice (Appendix A). These results suggest the significant role of PLC-β3 in IECs for interactions with microbiota and the more subtle roles played by PLC-β3 in other cell types. Interestingly, pathogenic *Escherichia albertii* and *Escherichia coli* were found in *Plcb3^−/−^* mice, and these pathogenic species increased after DSS treatment. Thus, assuming the clinical relevance of our mouse data in ileal CD, precision editing of the gut microbiota [85] could be beneficial for ileal CD patients. Further understanding of the PLC-β3-Wnt/β-catenin pathway will likely contribute to an appropriate selection of treatment for these patients.

## 5. Conclusions

In summary, we demonstrated that PLC-β3 in mammals and PLC-β in *Drosophila* regulate the Wnt/β-catenin signaling pathway at transcriptional, epigenetic, and, potentially, protein–protein interaction levels. A Wnt ligand binds to Fzd1 and then Fzd1/Gαq-bound PLC-β3 is activated, leading to the inactivation of the β-catenin destruction complex, followed by increased nuclear β-catenin, which activates the TCF/LEF transcription factors. Our data collectively suggest that the reduced activity of this evolutionally conserved PLC-β3-mediated Wnt/β-catenin signaling pathway contributes to the pathogenesis of ileal Crohn’s disease.

### Limitations of This Study

Our data suggest that PLC-β3 regulates the Wnt/β-catenin signaling pathway in the small intestine at transcriptional, epigenetic, and, probably, protein–protein interaction levels. We are aware of the weaknesses of this study. For example, some data were obtained with CMT-93 cells, which may be a less-than-ideal model as they express PLC-β3 more than normal cell counterparts. There are still gaps in our understanding of how Wnt/β-catenin signaling is controlled by PLC-β3, which is linked to the inactivation of β-catenin destruction complexes. Future studies will be needed to determine whether and how G proteins are involved in Wnt/β-catenin signaling, whether the enzymatic activity of PLC-β3 is required for Wnt/β-catenin signaling, and whether PLC-β3 regulates both Wnt/β-catenin and Wnt/Ca^2+^ signaling pathways. Furthermore, numerous protein–protein interactions detected by overexpression experiments need to be tested in more detail.

## Figures and Tables

**Figure 1 cells-13-00986-f001:**
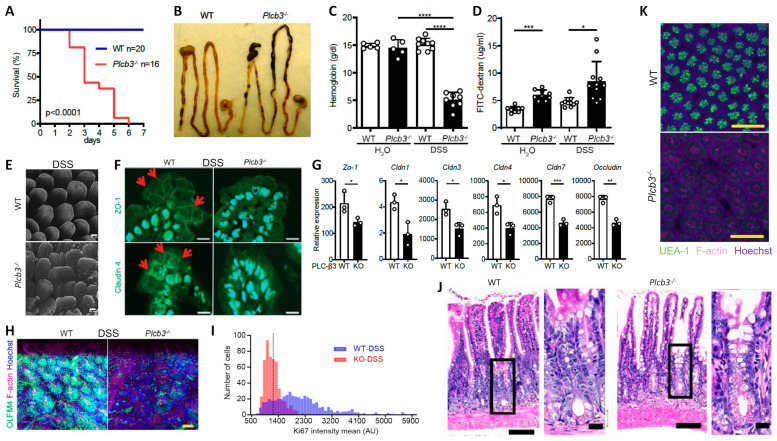
*Plcb3^−/−^* mice exhibited a DSS-induced lethal phenotype with severe inflammation in the small intestine and anemia. (**A**) Survival curves of mice exposed to 3% DSS for 5 days. (**B**) Gastrointestinal tracts of WT and *Plcb3^−/−^* mice after 1 day of exposure to 3% DSS. Note bloody stools in *Plcb3^−/−^* mice. (**C**) Hemoglobin concentrations in serum. **** *p* < 0.0001. (**D**) FITC-dextran was orally injected into mice pretreated with 3% DSS or H_2_O for 24 h; 6 h later, FITC-dextran in serum was quantified. The graph shows mean ± SD. * *p* < 0.05; *** *p* < 0.001, determined by Kruskal–Wallis test and Dunn’s multiple comparison test. (**E**) Whole mount staining of distal ilea of 2-day DSS-treated WT and *Plcb3^−/−^* mice were analyzed by confocal microscopy. Villi stained for F-actin by phalloidin, scale bar 50 μm. (**F**) Jejunal villi of WT and *Plcb3^−/−^* mice treated with DSS for 2 days were analyzed by confocal microscopy for ZO-1 (top) and claudin4 (bottom) expression. Arrows indicate staining at the cell–cell borders. Scale bar 10 μm. (**G**) qRT-PCR analysis of tight junction proteins in small intestines of DSS-untreated mice. * *p* < 0.05; ** *p* < 0.01 vs. WT by Student’s *t*-test. (**H**) Ileal crypts from mice treated with DSS for 2 days were stained by Hoechst (nuclei), anti-Olfm4 (ISCs), and phalloidin (F-actin). Scale bar 20 μm. (**I**) Histograms of Ki-67 mean fluorescence intensity (MFI) per crypt cells. In total, 500 ileal epithelial cells in DSS-treated mice were measured using sections stained for Ki-67. (**J**) H&E staining of ilea of WT or *Plcb3^−/−^* mice exposed to 3% DSS for 2 days. Portions indicated by rectangles are enlarged on the right. Scale bars 100 and 10 μm. (**K**) Whole mount ilea from mice treated with DSS for 2 days were stained by Hoechst, UEA-1, and phalloidin. Scale bar 100 μm.

**Figure 2 cells-13-00986-f002:**
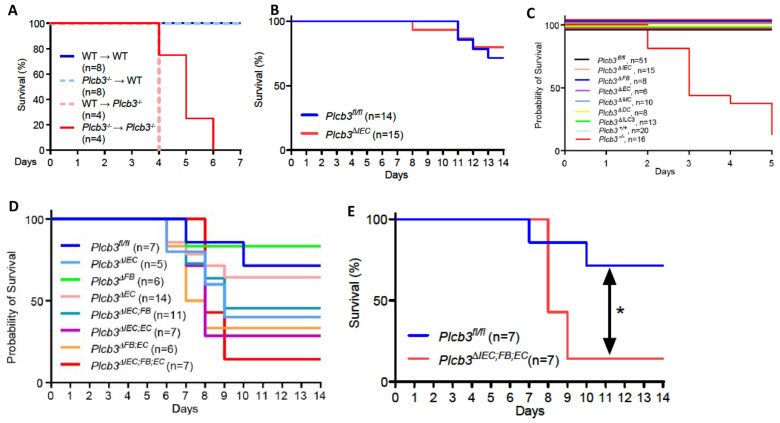
Multiple non-hematopoietic cell types contribute to the DSS-induced lethality of *Plcb3^−/−^* mice. (**A**) Survival curves of the recipient WT or *Plcb3^−/−^* mice that were orally treated with 3% DSS for 5 days 9 weeks after the bone marrow transfer. (**B**) Survival curves of *Plcb3^fl/fl^* and *Plcb3^fl/fl^;villin-Cre* (*Plcb3^ΔIEC^*) mice exposed to 3% DSS. (**C**,**D**) Survival curves of multiple conditional KO strains treated with 3% DSS. IEC: intestinal epithelial cell; FB: fibroblast; EC: endothelial cell; MC: myeloid cell; DC: dendritic cell; ILC3: type 3 innate lymphoid cell. (**E**) Survival curves of *Plcb3^ΔIEC;FB;EC^* mice or littermate *Plcb3^fl/fl^* control mice treated with 3% DSS. * *p* < 0.05 by log-rank test.

**Figure 3 cells-13-00986-f003:**
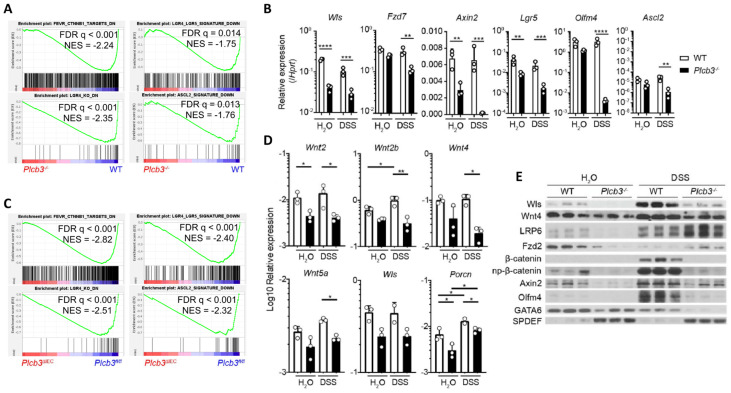
Small intestinal epithelial cells of *Plcb3^−/−^* mice exhibit downregulation of the Wnt/β-catenin signaling pathway. (**A**) GSEA indicates that the gene expression pattern in isolated IECs from *Plcb3^−/−^* mice correlates with the reduced expression of Wnt/β-catenin signature genes (upper left), Lgr4 regulated signature genes (lower left), Lgr4/Lgr5 regulated signature genes (upper right), and Ascl2 signature genes (lower right). NES: normalized enrichment score. *n* = 3 biological replicates. (**B**) qRT-PCR analysis of isolated IECs from WT and *Plcb3^−/−^* mice exposed to sterilized water or 3% DSS for 2 days. Normalized against *Hprt* mRNA. * *p* < 0.05; ** *p* < 0.01; *** *p* < 0.001; **** *p* < 0.0001, determined by two-tailed Student’s *t*-test. (**C**) GSEA with isolated IECs from *Plcb3 ^∆IEC^* mice indicated the reduced expression of Wnt/β-catenin (upper left), Lgr4 (lower left), Lgr4/Lgr5 (upper right), and Ascl2 (lower right) signature genes at the homeostatic condition. (**D**) Small intestines of WT and *Plcb3^−/−^* mice exposed to sterilized water or 3% DSS for 2 days were separated into mesenchymal tissues. RNAs isolated were subjected to qRT-PCR analysis. mRNA amounts for several Wnt ligands and Wnt secretion enablers (*Wls* and *Porcn*) were normalized against *Hprt*. * *p* < 0.05; ** *p* < 0.01, determined by Student’s *t*-test. (**E**) Western blot analysis of Wnt/β-catenin signaling proteins from WT and *Plcb3^−/−^* mice (3 mice each) under H_2_O and 2-day DSS treatments.

**Figure 4 cells-13-00986-f004:**
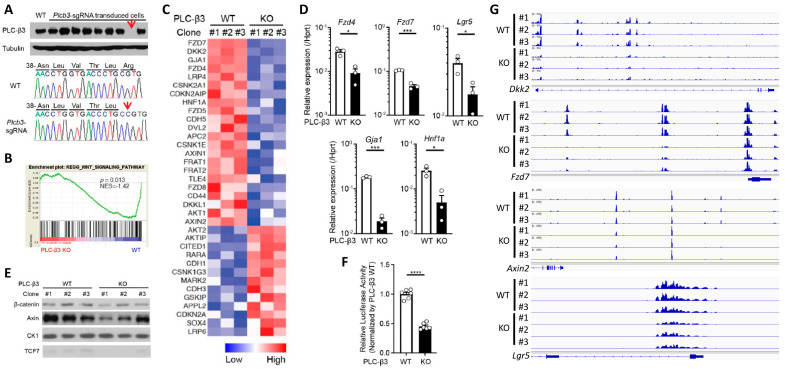
PLC-β3 regulates Wnt/β-catenin signaling at transcriptional, epigenetic, and protein–protein interaction levels in murine intestinal epithelial cells. (**A**) Western blot screening for *Plcb3*-deficient CMT-93 cells generated by CRISPR/Cas9. Only a small fraction of puromycin-selected lentivirus-transduced cells lacked PLC-β3 protein, probably due to the polyploidy of the cells. In clones #1 (shown) and #3 (not shown), a one-base insertion (indicated by an arrow) after Leu42-coding nucleotides in the *Plcb3* gene was confirmed by Sanger sequencing. Clone #2 (not shown) had a G insertion at the same position. The loss of PLC-β3 protein in the 3 clones was confirmed by Western blotting. (**B**) GSEA on *Plcb3*-deficient and -sufficient CMT93 cells. (**C**) Genes whose expression was most drastically altered by *Plcb3*-deficiency (22 down- and 13 up-regulated) found by IPA are listed. The order of listing is according to *p* values. (**D**) qRT-PCR analyses of several mRNAs in *Plcb3*-deficient and -sufficient CMT93 cells. (**E**) Western blot analysis of several Wnt/β-catenin signaling proteins. (**F**) TCF/LEF activity was analyzed by TOPFLASH luciferase assay using FOPFLASH as a negative control. **** *p* < 0.0001, determined by two-tailed Student’s *t*-test. (**G**) ATAC-seq analysis of *Plcb3*-deficient and -sufficient CMT-93 cells. (**H**–**J**) Co-immunoprecipitation of PLC-β3 with the indicated proteins. Mammalian expression vector with the indicated proteins as an insert or empty vector and PLC-β3 constructs or control plasmids were transfected to HEK293T cells. Anti-myc (**H**), anti-Dvl2 (I), or anti-Tcf-4 (**J**) antibodies were used for immunoprecipitation. IP: immunoprecipitation; IB: immunoblotting; IgG: control antibody. (**K**) CHO Chinese hamster ovary cells were co-transfected (TF) with vectors to express indicated proteins. The proximity of expressed proteins was visualized by proximity ligation assays (PLAs) using indicated pairs of primary antibodies. (**L**,**M**) WT and *Plcb3^−/−^* mice were treated with 0, 100, or 200 µg of R-spondin 1 for 24 h. IECs from these mice were analyzed by qRT-PCR (**L**) and Western blotting (**M**) for Wnt/β catenin signaling molecules and stem cell markers. mRNA expression was normalized against *Hprt*. * *p* < 0.05; ** *p* < 0.01; *** *p* < 0.001, determined by two-tailed Student’s *t*-test.

**Figure 5 cells-13-00986-f005:**
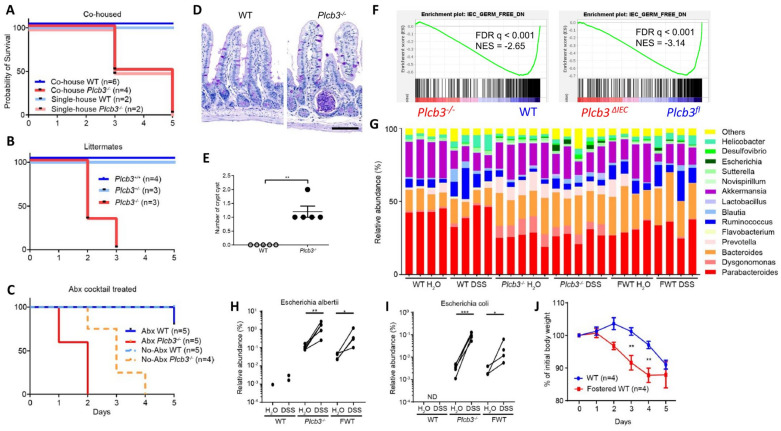
Dysbiosis contributes to the DSS susceptibility of PLC-β3 KO mice. (**A**–**C**) Survival curves of co-housed mice (**A**), littermates (**B**), or antibiotics cocktail (Abx)-treated (**C**) WT or *Plcb3^−/−^* mice. Mice were exposed to 3% DSS for 5 days. (**D**) PAS staining of ileum from WT or *Plcb3^−/−^* mice. Scale bar 100 μm. (**E**) Numbers of crypt abscesses in the whole intestine Swiss roll sections. ** *p* < 0.01, determined by Mann–Whitney test. (**F**) GSEA on isolated IECs from *Plcb3^−/−^* and *Plcb3^ΔIEC^* mice vs. their control mice with microbiota-induced genes. (**G**–**I**) 16S rRNA sequencing analysis of fecal pellets from WT, *Plcb3^−/−^*, and WT mice fostered by *Plcb3^−/−^* mice (FWT) before and after exposure to 3% DSS for 2 days. Genus-level data are shown (**G**). Relative abundance of *Escherichia albertii* (**H**) and *Escherichia coli* (**I**) are shown. * *p* < 0.05; ** *p* < 0.01; *** *p* < 0.001, determined by paired *t*-test. ND: not detected. (**J**) Body weight change in FWT and WT mice during exposure to 3% DSS for 5 days. FWT: WT mice fostered by *Plcb3^−/−^* mice. ** *p* < 0.01, determined by Sidak’s multiple comparison test.

**Figure 6 cells-13-00986-f006:**
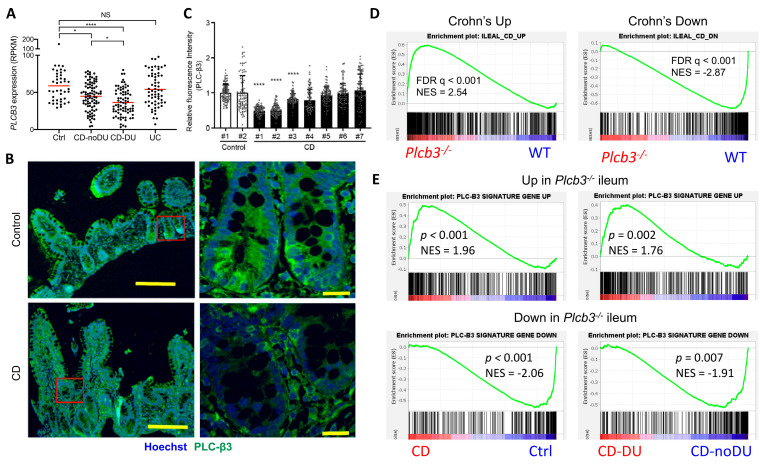
Reduced expression of PLC-β3 and altered expression of PLC-β3-regulated genes in ileal Crohn’s disease. (**A**) RNA-seq data on ileum biopsy samples from pediatric CD patients (GSE57945) were analyzed for *PLCB3* expression. Ctrl: control; CD: Crohn’s disease; DU: deep ulcer; UC: ulcerative colitis. (**B**,**C**) Biopsy samples from CD patients were stained for PLC-β3. (**B**) Representative images from 2 independent experiments. Scale bars 200 μm (left) and 20 μm (right). (**C**) Mean fluorescence intensity (MFI) for PLC-β3 from 100 epithelial cells was measured. MFI was normalized against control samples. * *p* < 0.05, **** *p* < 0.0001, determined by Kruskal–Wallis test and Dunn’s multiple comparison test. (**D**,**E**) Expression profiles of ilea from *Plcb3^−/−^* mice were analyzed by microarray. Then, 359 (upregulated) and 285 (downregulated) PLC-β3 signature genes were defined by ≥2-fold changes in expression and FDR q-values < 0.01 in *Plcb3^−/−^* mice. (**D**) GSEA was performed using RNA-seq data from the CD signature gene (GSE57945) and PLC-β3 signature gene sets. (**E**) GSEA was performed using RNA-seq data from CD patients (GSE57945) and the PLC-β3 signature gene set. A comparison between control and CD (left panels) or CD-No DU and CD-DU (right panels). NES: normalized enrichment score.

**Figure 7 cells-13-00986-f007:**
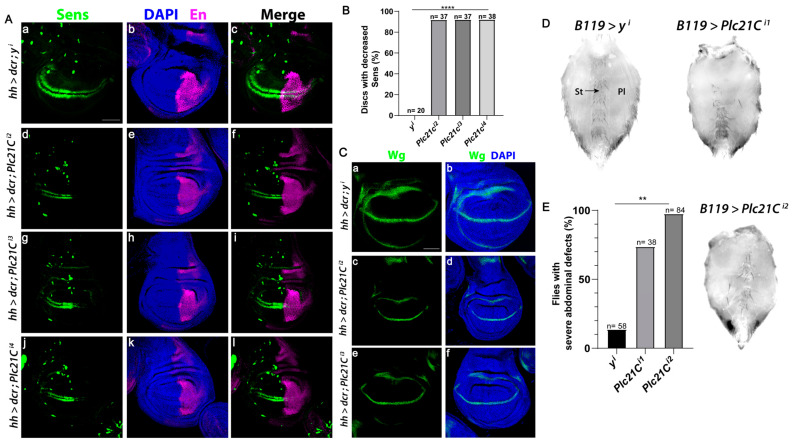
Plc21C is required for the expression of the Wingless target gene *Sens* in the Drosophila wing imaginal disc. (**A**–**C**) RNAi constructs targeting *Plc21C* or the *yellow (y)* negative control were expressed in the posterior compartment (marked by Engrailed (En)) of the third instar wing discs using the *hedgehog (hh)-Gal4* driver. A transcriptional reporter for *sens* expression, *mScar:T2A:Sens* (green), was analyzed. Scale bar (**a**–**l**) 20 µm. Dorsal, top and posterior, right. (**B**) Quantification is shown as a percentage of discs of each genotype with decreased Sens. N is the number of discs analyzed. (**C**) RNAi-mediated knockdown of Plc21C in the posterior compartment of third instar wing imaginal discs does not affect the expression of the *wingless* gene (green). Scale bar (**a**–**f**) 20 µm. Dorsal, top and posterior, right. (**D**,**E**) Loss of Plc21C in the Drosophila abdomen results in decreased Wingless signaling. Ventral abdomens of females expressing control RNAi targeting *y* using the *B119-Gal4* driver (top left) or those expressing RNAi constructs targeting *Plc21C* (top right and bottom right). Sternites and overlying sternal bristles (St, arrow) and pleura (Pl) are indicated. (**E**) At least 70% of all abdomens observed exhibited a severe phenotype. These abdomens were quantified as a percentage of total progeny. N is total number of abdomens of each genotype that were analyzed. ** *p* < 0.01; **** *p* < 0.0001.

## Data Availability

All unique/stable reagents generated in this study are available from the lead contact with a completed materials transfer agreement.

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
