# Peer review of "Ileal Crohn’s Disease Exhibits Reduced Activity of Phospholipase C-β3-Dependent Wnt/β-Catenin Signaling Pathway"

_cells, 2024, doi:10.3390/cells13110986_

Round 1

Reviewer 1 Report

Comments and Suggestions for Authors

Thank you for the opportunity to read and review this interesting manuscript. As a biostatistician I focused mainly on the methods of your article.

I have only one minor comment:

- I would have preferred it if all parts of the methods section were included in the article and not in the supplement

Author Response

Dear the reviewer,

Thank you for your positive comments.

We agree with you that the Methods should be included in the main text. I hope the editorial staff agrees with us.

Many thanks,

Toshiaki Kawakami

Reviewer 2 Report

Comments and Suggestions for Authors

The manuscript sought to investigate the role of phospholipase C-b3 in modulating signalling through the Wnt/b-catenin pathway during the pathogenesis of Crohn's disease (CD). Extensive and informative experiments were undertaken. Their well justified conclusions found that PLC-b3 regulates Wnt/b-catenin signalling through multiple regulatory mechanisms and promotes intestinal homeostasis and repair after injury. Thus, if PLC-b3 is reduced or absent, more severe damage will occur within the intestine following injury, such as occurs in Crohn's disease.

Importantly, the authors showed that PLC-3b is reduced in ileal biopsies from Crohn's disease patients, implicating a deficiency of PLC-3b in the pathogenesis of Crohn's. Notably, PLC-b3 regulation of Wnt/b-catenin signalling was also observed in Drosophila, suggesting strong evolutionary conservation of this mechanism.

The experiments undertaken were very extensive, utilising a range of PLC-b3 knock out mice, including conditional and cell-type specific knockouts, and tissue culture of specific mucosal cell types. From these extensive data the authors were able to demonstrate regulation by PLC-b3 within both hematopoietic and non-hematopoietic (mucosal) cells, at the transcriptional, epigenetic and likely protein-protein interaction levels.

The experiments were well designed and achieved appropriate statistical significance to justify the conclusions. The presentation of the data in text, figures and supplementary data was excellent. The authors provided an extensive discussion of their data, including relating their data to potential future therapeutic options for the treatment of Crohn's disease. The authors appropriately identified the limitations of their present studies and outlined future directions for investigation, particularly in relation to the extent and implications of likely protein-protein interactions during Wnt signalling.

Author Response

Dear the reviewer,

We appreciate your highly positive comments on our manuscript. Thank you very much for your time and effort in reviewing it.

Best regards,

Toshi

Reviewer 3 Report

Comments and Suggestions for Authors

After thorough review, I find the manuscript almost ready for acceptance as it is. No methodological issues to report. I’d like the authors to include in their review response answerers to the following questions:

Line 151; why have you used 5% DSS treatment protocol for the Plcb3dIEC mice ? Please justify the change.

Line 318; could you identify the strain of Escherichia coli in Plcb3-/- mice gut?

Further issues with manuscript formatting:

Line 130; keep Plcb3 -/- together and don’t allow the /- go to the next lane as it's currently.

Lines 139-162; correct the paragraph formatting

Author Response

Dear the reviewer,

Thank you very much for your highly positive comments on our manuscript. We also appreciate your time and effort to carefully read it.

Line 151: We tried 3% DSS in initial experiments with no prominent difference between WT and Plcb3-DeltaIEC mice. So, we have now changed this line to "...when fed 5%DSS (Figure S5E), but not 3% DSS (data niot shown) ..."

Line 318: Our identification of E. coli and other bacteria was totally dependent on 16S RNA-sequencing. Isolation and testing their role in determining DSS susceptibility.

Two other issues raised should be resolved by changes in formatting. I hope the editorial staff can do them.

Best regards,

Toshi